# On the Genuine Relevance of the Data-Driven Signal Decomposition-Based Multiscale Permutation Entropy

**DOI:** 10.3390/e24101343

**Published:** 2022-09-23

**Authors:** Meryem Jabloun, Philippe Ravier, Olivier Buttelli

**Affiliations:** Laboratoire Pluridisciplinaire de Recherche en Ingénierie des Systèmes, Mécanique, Énergétique (PRISME), University of Orleans, 45100 Orleans, France

**Keywords:** time series, multiscale permutation entropy, data-driven decomposition, nonlinear filtering, electromyography

## Abstract

Ordinal pattern-based approaches have great potential to capture intrinsic structures of dynamical systems, and therefore, they continue to be developed in various research fields. Among these, the permutation entropy (PE), defined as the Shannon entropy of ordinal probabilities, is an attractive time series complexity measure. Several multiscale variants (MPE) have been proposed in order to bring out hidden structures at different time scales. Multiscaling is achieved by combining linear or nonlinear preprocessing with PE calculation. However, the impact of such a preprocessing on the PE values is not fully characterized. In a previous study, we have theoretically decoupled the contribution of specific signal models to the PE values from that induced by the inner correlations of linear preprocessing filters. A variety of linear filters such as the autoregressive moving average (ARMA), Butterworth, and Chebyshev were tested. The current work is an extension to nonlinear preprocessing and especially to data-driven signal decomposition-based MPE. The empirical mode decomposition, variational mode decomposition, singular spectrum analysis-based decomposition and empirical wavelet transform are considered. We identify possible pitfalls in the interpretation of PE values induced by these nonlinear preprocessing, and hence, we contribute to improving the PE interpretation. The simulated dataset of representative processes such as white Gaussian noise, fractional Gaussian processes, ARMA models and synthetic sEMG signals as well as real-life sEMG signals are tested.

## 1. Introduction

Ordinal pattern-based approaches have recently gained attention because of their natural and efficient way to transform a time series into a sequence of symbols with a finite alphabet [1,2,3,4,5,6,7,8,9,10,11,12,13,14,15,16,17,18,19,20,21,22,23,24,25,26,27,28,29,30,31,32,33,34,35,36,37]. This ordinal pattern sequence is simply obtained by comparing the values of a finite number of neighboring samples and ranking them. This operation is threshold-free and does not require knowledge of the data range. Moreover, from a practical point of view, the obtained ordinal patterns are robust to linear perturbations and moderate noise levels [1,2]. The probability distribution of ordinal patterns (OP) implicitly captures information about the temporal structure of the time series and thus allows the extraction of several promising ordinal pattern-based indicators. These indicators are useful in a growing number of applications including biomedical signal and image processing [6,10,11,15,31,33,38], fault bearing diagnosis [30,39,40,41,42,43,44], financial time series analysis [7] and engineering physics [18,35,45,46,47,48,49,50,51].

Among the OP based-indicators, the permutation entropy (PE) was the first to be proposed [1]. It was introduced as a complexity measure of weakly stationary time series and was defined as the Shannon entropy [52] of the OP probability distribution. By construction, the PE benefits from the robustness and simplicity of the OP theory and takes into account information on the temporal organization of time series contrary to classical entropies. The PE is theoretically related to the well-known measure of dynamical systems complexity, the Kolmogorov–Sinai entropy [9]. A large number of time series processing tasks were performed using the PE tool [16,26,38,47].

As the PE has the ability to capture hidden dynamics of time series, various extensions of the PE, called multiscale PE (MPE), have been proposed to explore in depth the internal structure of signals at different time scales. Among the MPE techniques, we cite coarse-graining MPE [53], composite MPE [54], refined composite MPE [12], downspamling PE (DPE) [55], composite DPE [31], refined composite DPE [31], hierarchical PE [56], and data-driven decomposition-based MPE [38,39,40,41,42,43,44,45,46,47,48,49,50,51,57,58,59]. The main idea of these PE-based techniques is to combine PE with a preprocessing step of the signal of interest in order to reduce the number of patterns related to noise. These MPE techniques can be roughly split into two categories regarding the preprocessing step. The first category applies a linear filtering [12,31,53,54,55], whereas the second category uses a nonlinear processing [38,39,40,41,42,43,44,45,46,47,48,49,50,51,57,58,59]. Despite the widely proven usefulness of these techniques, there is still a need to understand the impact of the preprocessing step on the MPE values and how this impact can significantly alter the interpretation of MPE results. In [3], a study investigated the shortcomings of coarse-grained MPE, which can be viewed as a moving average filtering procedure followed by an M-factor downsampling operator. At high M-scales, this MPE suffers from spurious patterns reflecting spectral aliasing as Shannon’s theorem is no longer respected. In [27], we have theoretically characterized the effect of a linear filtering preprocessing on the PE. By using the Wiener–Khinchin theorem, the linear filter’s intrinsic PE and its contribution to the PE of the original signal were theoretically dissociated. This result was validated by means of simulated signals (white Gaussian noise and ARMA processes) subjected to a variety of linear filters such as the moving average, Butterworth, and Chebyshev type I filters. However, it is important to note that since the PE is a biased statistic [60], we have considered sufficiently long signals in order to neglect the bias effect. Therefore, the work presented in [27] provided an appropriate framework for characterizing the linear filter impact on the PE, which helped improve the interpretation of the PE by identifying possible artefact information introduced by this preprocessing step.

In the current paper, we aim at characterizing the data-driven decomposition-based MPE as a new challenge in the understanding of the impact of nonlinear preprocessing in the PE. To that end, four decomposition methods are considered: the empirical mode decomposition (EMD) [39,40,43,48,57], the variational mode decomposition (VMD) [38,41,45,46,47,49,58], the singular spectrum signal analysis-based decomposition (SSA) [50,59] and the empirical wavelet transform [42,44,51]. We test a variety of simulated signals including white Gaussian noise, fractional Gaussian noise, ARMA processes, simulated surface electromyography signals (sEMG) and multicomponent sinusoidal signals. We find that the multiscaling effect of these methods should be interpreted with respect to the mean frequency of the components obtained and not as a function of the rank of these obtained components. The observed decrease of the PE of each component is not related to changes on the PE but only to the spectrum shift of those components. This may invalidate some interpretation of the results of such tools, especially when applied to real data.

The paper is organized as follows. Section 2 recalls the theoretical background of PE. Section 3 elaborates on the data-driven decomposition-based MPE and the results that bring light on the MPE are presented. Discussion is provided in Section 4. Section 5 draws conclusions and arguably.

## 2. Permutation Entropy—Review of Existing Theory

This section briefly recalls the concepts and tools useful for understanding the PE and MPE.

### 2.1. Permutation Entropy Definition

Let us consider *N* samples of a weakly stationary and uniformly sampled time series $x_{t}$ with t=0,1,…,N−1. A symbol sequence, called pattern $\Pi \;=\;r_{1}r_{2}\ldots r_{d}$ of dimension (order) *d*, can be obtained by sorting in ascending order every *d* consecutive samples $x_{t},x_{t+1},x_{t+2},\ldots ,x_{t+d-1}$. This can be represented by a permutation function v(.) of the set $\left\{1,2,\ldots ,d\right\}$ defined by: $v\left(1\right)=r_{1}$, $v\left(2\right)=r_{2}$, $v\left(3\right)=r_{3},\ldots$, and $v\left(d\right)=r_{d}$. The maximum number of permutation of this set is d!, and so is the maximal number of distinct patterns.

For example, for d=3, if $x_{t}<x_{t+1}<x_{t+2}$, we obtain the pattern Π=123 with $r_{1}=1$, $r_{2}=2$, $r_{3}=3$ and v(.) is the identity, whereas if $x_{t+1}<x_{t}<x_{t+2}$, we obtain the pattern Π=213 where $r_{1}=2$, $r_{2}=1$ and $r_{3}=3$.

The probability that the pattern Π of type $r_{1}r_{2}\;\ldots \;r_{d}$ occurs can be estimated [2], subject to N>>>d! by
(1)$$p_{r_{1}r_{2}\;\ldots \;r_{d}}=\left\{\frac{\#t\;\;|\left\{x_{t},x_{t+1},\ldots ,x_{t+d-1}\right\}\;\mathrm{of}\;\mathrm{type}\;r_{1}r_{2}\ldots r_{d}}{N-d-1}\right\},$$
where # denotes the cardinal number.

The normalized PE, which was introduced in [1] as a natural measure of the complexity of the signal $x_{t}$, was defined as the Shannon entropy applied to the pattern distribution (Equation 1), and it can be estimated as:(2)$$H=-\frac{\sum p_{\Pi }\log\left(p_{\Pi }\right)}{\log(d!)}.$$
The PE reaches its minimum value 0 for a monotone waveform, whereas it reaches its maximum value of 1 for a uniform pattern distribution ($p_{\Pi_{i}}=\frac{1}{6}$ for d=3 and $p_{\Pi_{i}}=\frac{1}{24}$ for d=4). The PE robustness to ectopic or aberrant values and to moderate noise level is well established [60,61,62].

### 2.2. Permutation Entropy of Known Signals

Several studies have been conducted on the properties and behavior of PE on known signals [1,2,7,55,60,62,63,64]. For a comparison purpose, we here recall the results obtained on stationary real Gaussian processes and pure tone signals.

#### 2.2.1. Stationary Real Gaussian Processes

In [2], the PE (Equation 2) of a stationary real Gaussian process $x_{t}$ of zero mean and unit variance was evaluated using the exact expressions of the probability of ordinal patterns $\Pi =r_{1}r_{2}\;\ldots \;r_{d}$ provided that 2≤d≤4. We propose to merge all these expressions into one written as follows: (3)$$\begin{array}{cc} \; & p_{r_{1}r_{2}\ldots r_{d}}=\frac{1}{2^{d-1}}+\frac{1}{2^{d-2}\pi }\;\;\sum_{\underset{i<j}{i,j\in \left\{1,2,\ldots ,d-1\right\}}}\;\arcsin\;\left(\frac{1}{2}\frac{\alpha_{i,j}}{\beta_{i,j}}\right) \end{array}$$(4)$$\begin{array}{cc} \mathrm{where}\; & \alpha_{i,j}=\sum_{\ell ,h\in \left\{0,1\right\}}{\left(-1\right)}^{\ell +h}\rho_{xx}\left(v(i+\ell )-v(j+h)\right) \end{array}$$(5)$$\begin{array}{cc} \; & \beta_{i,j}^{2}=\left(1-\rho_{xx}\left(v(i+1)-v(i)\right)\right)\left(1-\rho_{xx}\left(v(j+1)-v(j)\right)\right), \end{array}$$
and $\rho_{xx}\left(\tau \right)$ denotes the normalized auto-correlation function (ACF) of the real time series $x_{t}$ defined as:(6)$$\rho_{xx}\left(\tau \right)=\rho_{xx_{\tau }}=\frac{E\left[x_{t}\;\;x_{t+\tau }\right]}{E[x_{t}^{2}]}$$
for τ=−N+1,−N+2,…,N−1. Note that $\rho_{xx_{\tau }}$ is symmetric and that both () and () only depend on the values of $\rho_{xx_{1}}$, $\rho_{xx_{2}}$ and $\rho_{xx_{3}}$.

Thus, knowing these three point values of the ACF of any stationary Gaussian signal is sufficient to access to its corresponding theoretical PE value.

Among the stationary Gaussian processes considered in [2,60,62], there are

Autoregressive AR(1) processes defined by $x_{t}=a\;x_{t-1}+\epsilon_{t}$ where the parameter |a|<1, and $\epsilon_{t}$ is an i.i.d. zero-mean and unit variance Gaussian noise. Its ACF is given by
(7)$$\rho_{xx_{\tau }}=\left\{\begin{array}{cc} 1 & \;\mathrm{if}\;\tau =0,\\ a^{\left|\tau \right|} & otherwise. \end{array}\right.$$AR(2) processes defined by $x_{t}=ax_{t-1}+bx_{t-2}+\epsilon_{t}$ with parameters *a* and *b* verifying |a+b|<1, |a−b|<1 and |b|<1. Its ACF is calculated through
(8)$$\rho_{xx_{\tau }}=\left\{\begin{array}{cc} 1 & \;\mathrm{if}\;\tau =0,\\ \frac{a}{1-b} & \;\mathrm{if}\;\tau =1,\\ a\rho_{xx_{\tau -1}}+b\rho_{xx_{\tau -2}} & otherwise. \end{array}\right.$$Moving average processes MA(*q*) defined by $x_{t}=\sum_{\ell =0}^{q}a_{\ell }\epsilon_{t-\ell }$ with $a_{0}=1$, and $\epsilon_{t}$ is a unitary centered Gaussian noise. Its ACF is evaluated using the following expression:
(9)$$\rho_{xx_{\tau }}=\left\{\begin{array}{cc} \frac{\sum_{\ell =\tau }^{q}a_{\ell }a_{\ell -\tau }}{\sum_{\ell =0}^{q}a_{\ell }^{2}} & \;\mathrm{if}\;|\tau |\leq q,\\ 0 & otherwise. \end{array}\right.$$Fractional Gaussian noise (fGn) is defined as the increment of fractional Brownian motion (fBm) and is a zero-mean stationary process. Its ACF is given by
(10)$$\rho_{xx_{\tau }}=\frac{1}{2}\left({\left|\tau +1\right|}^{2h}-{2|\tau |}^{2h}+{\left|\tau -1\right|}^{2h}\right).$$
where h∈]0,1[ is referred to as the Hurst exponent [55]. Recall that for h=0.5, the fGn reduces to a white Gaussian noise with equal probabilities for the d! order patterns of dimension *d* and hence a permutation entropy *H* = 1.Synthetic sEMG signals simulated as Gaussian processes, whose power spectral density (PSD) is modeled according to [65] by:
(11)$$PSD\left(\nu \right)=k\frac{f_{h}^{4}\nu^{2}}{\left(\nu^{2}+f_{l}^{2}\right){\left(\nu^{2}+f_{h}^{2}\right)}^{2}}$$
where *k* is a normalization constant and $f_{l}$ and $f_{h}$ are the normalized low and high cut-off frequencies of the sEMG PSD model. Its ACF expression was derived in [27]:
(12)$$\rho_{xx_{\tau }}=\frac{2f_{h}}{{\left(f_{h}-f_{l}\right)}^{2}}\left(-f_{l}\;\;e^{-2\pi f_{l}\left|\tau \right|}+\frac{f_{h}^{2}+f_{l}^{2}}{2f_{h}}e^{-2\pi f_{h}\left|\tau \right|}+\pi \left(f_{h}^{2}-f_{l}^{2}\right)\;\left|\tau \right|e^{-2\pi f_{h}\left|\tau \right|}\right).$$Correlated Gaussian processes defined as the output of a linear filter whose input is a white Gaussian noise. Such processes can be encountered in studies limited to one specific or useful bandwidth as it is the case in [66]. The ACF of such processes is directly linked to the ACF of the used linear filter as described in [27]. For example, an ideal band-pass filter of a normalized bandwidth is equal to $BW=[\nu_{l},\nu_{f}]$, whose transfer function is defined as:
(13)$$HF\left(\nu \right)=\left\{\begin{array}{cc} 1 & \mathrm{if}\;\nu \in [\nu_{l},\nu_{h}],\\ 0 & otherwise \end{array}\right.,$$
with $0\leq \nu_{l}<\nu_{h}<0.5$ and ν is the normalized frequency, which leads to an ACF of the filter output signal as follows:
(14)$$\rho_{xx_{\tau }}=\cos\left(\pi (\nu_{h}+\nu_{l})\tau \right)\;\sin c\left(\pi (\nu_{h}-\nu_{l})\tau \right).$$Note that by setting $\nu_{l}=0$, we have the ACF of the output signal of ideal low-pass filters. Another example, a Gaussian filter whose transfer function is defined by
(15)$$HF\left(\nu \right)=\exp\left(-2\pi \nu^{2}\sigma^{2}\right)$$
leads to an ACF of the following filter output signal
(16)$$\rho_{xx_{\tau }}=\exp\left(\frac{-\tau^{2}}{2\sigma^{2}}\right).$$

#### 2.2.2. Sinusoidal Signals

Let us consider a pure tone signal $x_{t}=\cos\left(2\pi \nu_{0}t\right)$ with a normalized frequency $\nu_{0}=\frac{f_{0}}{F_{s}}$ and a large number of samples in order to avoid bias estimation of the PE [60]. The PE value can be approximated by the following expressions for $F_{s}>>>2f_{0}$:(17)$$\begin{array}{cc} H=-\frac{2\nu_{p}\log\left(\nu_{p}\right)+\left(1-2\nu_{p}\right)\log\left(\frac{1}{2}-\nu_{p}\right)}{\log(3!)} & \;\mathrm{for}\;d=3, \end{array}$$(18)$$\begin{array}{cc} H=-\frac{4\nu_{p}\log\left(\nu_{p}\right)+\left(1-4\nu_{p}\right)\log\left(\frac{1}{2}-2\nu_{p}\right)}{\log(4!)} & \;\mathrm{for}\;d=4, \end{array}$$(19)$$\begin{array}{c} \mathrm{with}\;\nu_{p}=\frac{\nu_{0}}{1-\nu_{0}\left(d-1\right)}. \end{array}$$
When the sampling frequency $F_{s}$ tends to be very high in comparison to the signal frequency $f_{0}$, i.e., $\nu_{p}\approx 0$, the PE *H* reaches the theoretical PE of a continuous time series H=0.387 and H=0.218 for d=3 and d=4, respectively [67]. It turns out that the PE value H=0.387 is also that obtained for d=3 when the sampling frequency is equal to the Nyquist rate $F_{s}=2f_{0}$. Indeed, in this last case, only two equiprobable patterns are available.

Now, let us consider a sum of two sinusoidal signals $x_{t}=\cos\left(2\pi \nu_{0}t\right)+a_{r}\cos\left(2\pi f_{r}\nu_{0}t+\phi \right)$, where parameters $a_{r}$ and $f_{r}$ are the amplitude and frequency ratios of the two considered components, and ϕ is a uniform random phase identically distributed on [−π,π]. The average of this signal PE, <H>${}_{\phi }$, with respect to ϕ is a function of the parameters $a_{r}$ and $f_{r}$ as shown in Figure 1a,b, where $f_{0}=1$ Hz and $F_{s}=20$ Hz. The dark area corresponds to a PE close to that of the dominant component $\cos(2\pi \nu_{0}t)$ and equal to 0.581 and 0.441 for d=3 and d=4, respectively.

Note that this area edge and the PE values are sensitive to the chosen sampling frequency [67] and to the signal-to-noise ratio (SNR), as illustrated in Figure 2a,b.

### 2.3. Multiscale Permutation Entropy

Multiscale variants of PE (MPE) have been proposed in the literature in order to identify hidden structures in long memory processes or/and over-sampled signals. The main idea of MPE methods is to apply a preprocessing step to the signal of interest in order to reduce the number of noise-related patterns before proceeding to PE estimation.

#### 2.3.1. Linear Preprocessing-Based MPE

Many of the MPE are based on linear preprocessing [12,30,31,53,54,55] such as lag operator and/or linear filters and/or downsampling. Firstly, it should be noted that at high scales, the downsampling step of this MPE class may induce spurious patterns reflecting spectral aliasing as Shannon’s theorem is no longer respected [3]. Secondly, the constraint $\frac{N}{M}>>d!$, where *N* is the signal sample number and *M* is the scale, must be verified in order to ensure a reduced bias and variance of the estimate of the pattern distribution (Equation 13) [60,62]. Finally, the linear filtering has an impact on the PE, which has been theoretically and experimentally characterized in [27]. More precisely, in [27], the contribution of the linear filter to the PE was theoretically dissociated from that of the original signal model. Therefore, the work presented in [27] helped improve the interpretation of the PE by identifying possible artefact information introduced by this linear preprocessing step.

#### 2.3.2. Nonlinear Preprocessing-Based MPE

Other MPE methods that use a nonlinear preprocessing have been proposed in order to overcome the shortcomings of the aforementioned linear filtering-based MPE class [38,39,40,41,42,43,44,45,46,47,48,49,50,51,57,58,59].

Among these nonlinear preprocessing-based MPE, we cited four data-driven signal decomposition-based MPE that are of great interest in many engineering fields: the empirical mode decomposition (EMD) [39,40,43,48,57], the variational mode decomposition (VMD) [38,41,45,46,47,49,58], the singular spectrum signal analysis-based decomposition (SSA) [50,59] and the empirical wavelet transform [42,44,51].

The EMD is a well-known data-driven approach that was proposed by Huang et al. [68] to recursively decompose a signal into a sum of oscillating functions called intrinsic mode functions (IMF). The empirical scheme used by the EMD is based on a sifting process that iteratively extracts the envelops of signal maxima and minima and removes their mean from the signal. An IMF is extracted when a stopping criterion of the sifting process is reached. The original stopping criterion was defined as the total number of extrema and the number of zero-crossings differs by at most one. The obtained IMFs should have separated spectral bands and be well-behaved with the Hilbert transform. The aspect of the EMD acting as a filter bank has been also studied in depth in [69]. In practice, the EMD is known to be sensitive to noise, sampling frequency and mode mixing of signal components with close frequencies. Other considerations or/and modifications of the original EMD have also been proposed to overcome the EMD limitations [70,71,72]. In our study, we use the EMD implementation proposed in [70] where a Cauchy-type stop criterion is considered in order to relax the original IMF definition and to extract AM-FM modes with physical significance.

The VMD proposed in [73] is a non-recursive signal decomposition method that breaks down the signal into a small number of narrow-band IMFs well adapted to Hilbert transform. Unlike EMD, the VMD has a consolidated mathematical theory, since the IMFs and their respective center frequencies are extracted concurrently by optimizing a constrained variational problem. All IMFs obtained are amplitude and frequency-modulated (AM-FM) cosine waveforms whose instantaneous frequencies vary slowly in a non-decreasing pattern. The VMD exhibits interesting performances compared to the EMD, such as the attenuation of the mixing mode and the improvement of the noise robustness. The VMD behavior as a filter bank was investigated in [74]. In order to increase the performance of the VMD, improved versions of the VMD were also proposed [75,76,77].

The empirical wavelet transform (EWT) proposed in [78] is a data-driven decomposition method that extracts different oscillatory modes from a signal by designing an adaptive wavelet filter bank. The upper and lower passband cut-off frequencies of the wavelet filters are automatically selected as the midpoint of two adjacent local maxima of the Fourier spectrum of the signal. The extracted modes are narrow time-frequency sub-bands. As with VMD, the EWT has a theoretical basis. It also combines the advantages of EMD and the wavelet transform. Further refinements of EWT have also been proposed in [79,80,81] to improve the performance of EWT.

Singular spectrum analysis (SSA), introduced in [82] and featuring an unsupervised classification algorithm, is also a data-driven decomposition method, but it differs in that its theoretical basis is close to that of the principal component analysis. SSA is able to identify principal processes and hidden periodicities. The components extracted are physically significant and correspond either to orthogonal oscillatory components with a narrow spectral band (periodic or quasi-periodic components), trends or noise. Various improvements to the original SSA have also been proposed to overcome the redundancy of the original SSA decomposition and to take into account the possibility of a time-varying number of components [83,84].

Despite the widely proven usefulness of the permutation entropy combined with these data-driven decomposition-based techniques, there is still a need to understand the impact of such a preprocessing on the MPE values and how this impact can significantly alter the interpretation of the MPE results.

## 3. Impact of Nonlinear Preprocessing in the PE Values

In this section, we aim to characterize data-driven decomposition-based MPE as a new challenge in understanding the impact of nonlinear preprocessing in PE. We compare the four signal decomposition-based methods: EMD, VMD, EWT and SSA.

We present results using simulations of different models, including white noise, fractional Gaussian noise (fGn), AR(q) and MA(q) models, and simulated sEMG signals. An application to real-life sEMG data is also provided.

For all curves, the markers indicate the actual calculated PE values of extracted components (IMFs), while the lines are an interpolation of these values.

### 3.1. Results on White Noise

A total of 100 time series of uncorrelated white Gaussian noise were simulated. The sampling frequency was set to $F_{s}$ = 1000 Hz and the number of samples was equal to N=1000. At least eight components (IMFs) were extracted from each time series and for each method. The PE was then calculated for each obtained component. We also propose to compute the normalized center frequency of each component for comparison purposes. This can be achieved using a PSD estimation, PSD(ν), of the IMF:(20)$$\nu_{c}=\int \nu \;PSD\left(\nu \right)\;d\nu .$$

Figure 3 shows a comparison of the PE obtained from simulations. For all considered methods, the results are presented in two ways. The first one is the classical one: PE as a function of the extraction rank of the components. In the second way, PE is presented as a function of the normalized center frequency of the IMFs.

As can be seen in Figure 3a,c, the four methods considered seem to act differently on the PE, but looking at Figure 3b,d, the variation and decrease of the PE with respect to the center frequency of extracted components seem to be similar for almost all methods. The results are very close especially for the components extracted at high or moderate frequencies with VMD and EWT preprocessing methods. The SSA method sticks to a single spectral band, and the first 25 extracted components all share the same narrow band. The lowest number of extracted components is obtained by EMD, but it covers a larger spectral range.

Note that these obtained results remain valid even if the uncorrelated noise is not Gaussian but uniformly distributed.

Next, to further investigate the impact on the MPE and to obtain an idea of the reliability of these results, we present a theoretical approximation of the EMD and VMD-based MPE. To this end, recall that both the EMD and VMD act as a filter bank but in two distinct ways [69,74]. Figure 4a,d illustrates this filter-bank aspect where the DSPs of the first eight IMFs extracted from Gaussian noise are depicted. We propose to approximate their normalized PSD by Gaussian curves:(21)$$PSD\left(\nu \right)\approx \exp(-4\pi^{2}{\left(\nu -\nu_{c}\right)}^{2}\sigma^{2}).$$
From this point, each IMF can be viewed as the output of a linear filter whose input is white Gaussian noise and transfer function is the square root of the expected spectra (Equation 21). An approximation of the normalized ACF is then given by (Equation 16), and an approximation of the theoretical PE of the IMF can be obtained using (Equation 3). The theoretical PE obtained are shown in Figure 4b,e for a pattern dimension d=3 and in Figure 4c,f for a pattern dimension d=4. As shown in these figures, the theoretical and experimental PE appear to have a close shape, especially at high and moderate center frequencies. At very low center frequencies, the Gaussian approximation of the PSD IMF may no longer be valid, leading to a divergence between the experimental and theoretical PE.

### 3.2. Results on Fractional Gaussian Noise

The application of the data-driven decomposition-methods (EMD and VMD) on fractional Gaussian noise (fGn) is illustrated in Figure 5 and Figure 6. In total, 100 simulated time series with a sample number N=1000 were generated for each Hurst exponent.

As can be seen in Figure 5a,c, the two methods considered appear to act differently on the fGn PE values. In addition, the fGn with the long-range dependence (h=0.9) has the lowest IMF PE values.

However, as shown in Figure 5b,d and Figure 6a, the decay and variation of the IMF PE curves with respect to the IMF center frequencies seem to be very close or even identical to those of the white Gaussian noise (h=0.5), especially for IMFs of high and moderate center frequencies. Furthermore, Figure 6a shows that the shape of the IMF PE curves is non-sensitive to the Hurst exponent, but the actually calculated PE values that are indicated by the (o) marker are, however, sensitive to a frequency shift.

Therefore, if we model the PE curve of AWGN IMFs by a function $H=\mathrm{Fct}\left(\nu_{c}\right)$ of the normalized center frequency $\nu_{c}$, we can easily obtain the PE of fGN IMFs by scaling: $H_{fGn}=H\left(\alpha_{h}\nu_{c}\right)$, where $\alpha_{h}$ is a slope that depends on the Hurst exponent. These slopes are actually depicted in Figure 6b where we can see the linear relationship between normalized center frequencies of the fGn IMFs and those of AWGN IMFs.

### 3.3. Results on Signal Models

We compute the PE of correlated signals in order to assess the contribution of the data-driven decomposition method to the original PE of the signal models. We use white Gaussian noise as a benchmark for comparison. We simulate 100 time series of length N=1000 for each of the following models.

AR(2) process with a = 0.8 and b = 0.15 and whose PE is evaluated by (Equation 8) and (Equation 3) [27].AR(7) processes used in [85] to model the PSD of Heart Rate Variability (HRV) recorded in the tilt and rest positions. Table III in [85] shows the AR model parameters used for a sampling rate of $F_{s}=$1 Hz and error term variances equal to 404 × 10${}^{-6}$ and 137 ×$10^{-6}$ for rest and tilt, respectively.MA(2) process with parameters $a_{0}=0.4$ and $a_{1}=0.6$ and whose PE is evaluated by (Equation 9) and (Equation 3) [27].sEMG models: sEMG signals are obtained by filtering white Gaussian noise with the inverse Fourier transform of the square root of expected spectra (Equation 11). Two sets of parameters $\left(f_{l},f_{h}\right)$ are tested: (30 Hz, 60 Hz) and (50 Hz, 150 Hz) for a sampling frequency $F_{s}=1024$ Hz as proposed in [65]. The PE of such signals is evaluated using (Equation 12) and (Equation 3) [27].

We also propose theoretically investigating the impact of the EMD-based MPE on the contribution of those models by using the same theoretical Gaussian approximation of IMFs DSP, as detailed in Section 3.1.

The application of EMD-based MPE to the signal models considered above is illustrated in Figure 7. The comparison between the PE of AR(2) IMFs and the AWGN IMFs (see Figure 7e) shows superimposed PE curves, but the actually calculated PE values that are indicated by the (o) marker are shifted. So, we notice the same PE scaling effect with respect to the center frequencies of IMFs as in the case of fGn. This remark also holds for the MA(2) model, as shown in Figure 7d, and for the IMF PE curves of the AR(7) models of HRV in tilt and rest position, as shown in Figure 7c. The IMF PE of simulated sEMG signals also exhibits the same behavior (PE scaling with respect to center frequency), as shown in Figure 7e. However, we notice that this latter figure shows that the highest frequency extracted components are slightly sensitive to the signal model.

In Figure 7b, we have also added the results of a linear filtering on the PE values where, unlike EMD-based MPE, there is a gap between the filter intrinsic PE and the AR(2) signal model filtered at high frequencies.

Finally, we propose to check the impact of the data-driven decomposition on the PE values of the sum of two sinusoidal signals $x_{t}=\cos\left(2\pi \nu_{0}t\right)+a_{r}\cos\left(2\pi f_{r}\nu_{0}t+\phi \right)$ as a function of the parameters $a_{r}$ and fr.

To this end, we calculate the relative percentage error between the theoretical PE (Equation 17) and the computed PE of pattern dimension d=3 for the first extracted component. This error is plotted in Figure 8 as a function of the amplitude and frequency ratios. As can be seen in Figure 8, the error variation in PE values reflects the ability of the methods considered (EMD, VMD, EWT and SSA) to separate the components.

### 3.4. Results on Real-Life sEMG Signals

We here consider two sEMG signals recorded from the short head of the biceps brachii during exercises according to the protocol detailed in [86]. The force level was imposed at 20% and at 80%. From Figure 9, we notice that when plotted against the center frequencies of the IMFs, the EMD or VMD-based MPE is very close to that of white Gaussian noise. Only the lowest frequency IMFs have discarded PE. The most noticeable effect is again the frequency shift or the PE scaling effect, as shown in Figure 9c,f.

## 4. Discussion

The data-driven decomposition-based methods we considered extract components with different frequency panels. When applied to AWGN or to correlated processes with different signal models (AR, MA, simulated sEMG), all the considered methods behaved almost identically to a white Gaussian noise with respect to the center frequencies of the extracted components. Indeed, the PE values of extracted components are located on the same curve used as a benchmark whatever the model and for each of the data-driven decomposition methods used. However, the actually calculated PE values which are of the same number of extracted IMFs seem to be shifted frequency with respect to those of AWGN.

Unlike linear preprocessing MPE, it can be concluded that the variation in the data-driven decomposition-based MPE may mainly reflect the frequency shift (frequency bandwidth of extracted components) of components more than a concrete complexity variation related to the signal model. The obtained results suggest that the PE decay or variation should not be interpreted from the perspective of rank of the extracted component but rather from the perspective of their frequency location. These decay and variation are sensitive to the frequency shift of extracted components.

## 5. Conclusions

We need to accurately outline the impact of nonlinear preprocessing in the PE and identify its contribution to PE from the information content of signal models. In this work, we found that data-driven decomposition-based MPE should be analyzed and interpreted based on the center frequency of the extracted components rather than their extraction rank. Unlike the linear filtering-based MPE, the variations of these data-driven decomposition-based MPE mainly reflect the frequency shift of the bandwidth of each IMF more than the contribution of the signal model. This work allows us to correctly interpret the PE values over an arbitrary time series. Future research directions may also involve highlighting the impact in special cases such as existing persistent forbidden patterns. This is a topic for future research to explore.

## Figures and Tables

**Figure 1 entropy-24-01343-f001:**
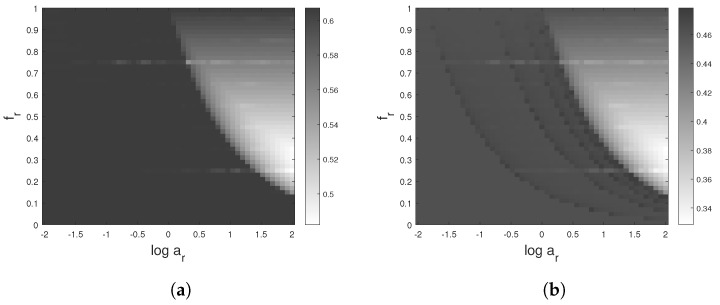
PE values of the sum of two sinusoidal signals $x_{t}=\cos\left(2\pi \nu_{0}t\right)+a_{r}\cos\left(2\pi f_{r}\nu_{0}t+\phi \right)$ as a function of the parameters $a_{r}$ and $f_{r}$. The signal is noise free, and the sampling frequency is Fs=20 Hz. (**a**) d = 3; (**b**) d = 4.

**Figure 2 entropy-24-01343-f002:**
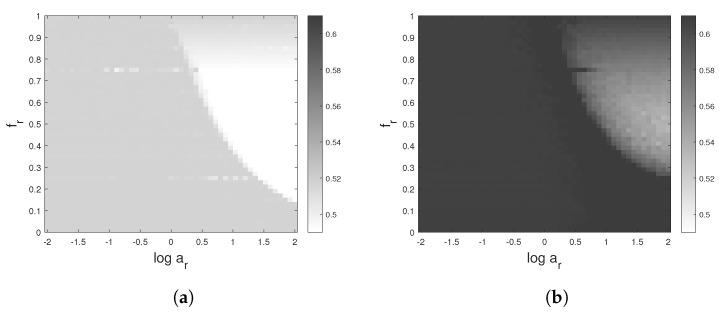
PE values of the sum of two sinusoids (Equation 17) as a function of the parameters $a_{r}$ and $f_{r}$: PE values are dependent of the sampling frequency and the SNR. (**a**) d=3, $F_{s}=40$ Hz and SNR=*∞*; (**b**) d=3, $F_{s}=20$ Hz and SNR = 20 dB.

**Figure 3 entropy-24-01343-f003:**
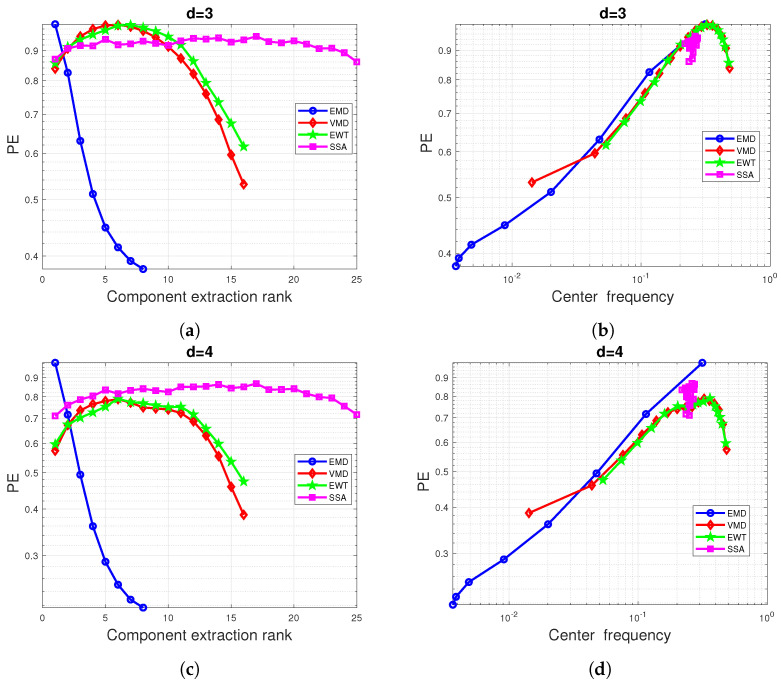
Impact of nonlinear preprocessing on the MPE values of white Gaussian noise: EMD, VMD, EWT or SSA decomposition precedes the calculation of PE with a pattern dimension d=3 and d=4. (**a**,**c**) PE as a function of the extraction rank of the components in contrast to (**b**,**d**) PE as a function of the center frequency of each extracted mode. The average results of 100 simulated time series are depicted. At least 8 components are extracted by each method.

**Figure 4 entropy-24-01343-f004:**
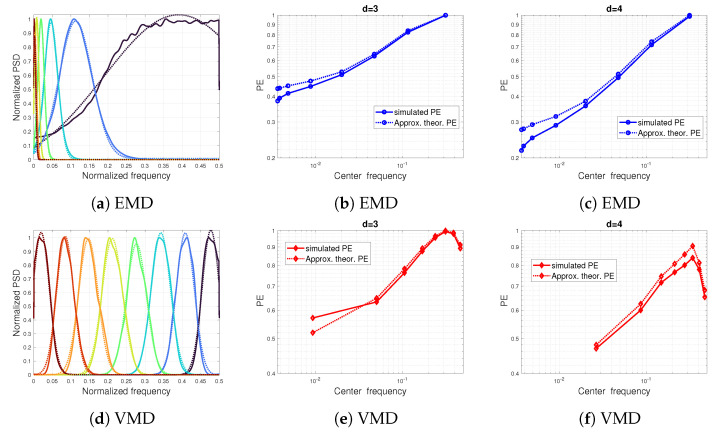
Impact of nonlinear preprocessing on PE values of white Gaussian noise: EMD (or VMD) precedes the calculation of PEs with a pattern dimension d=3 and d=4. (**a**,**d**) Estimation of the IMF PSDs (solid line) and the Gaussian approximation of these PSD (dotted line). (**b**,**e**) Average of PEs of the extracted IMFs based on 100 simulated Gaussian white noise superimposed to their approximated theoretical PEs computed using (Equation 16) and (Equation 2) for a pattern dimension d=3. (**c**,**f**) same as (**b**,**c**) but with a pattern dimension d=4. All calculated PEs are plotted against the average of center frequencies of each extracted mode.

**Figure 5 entropy-24-01343-f005:**
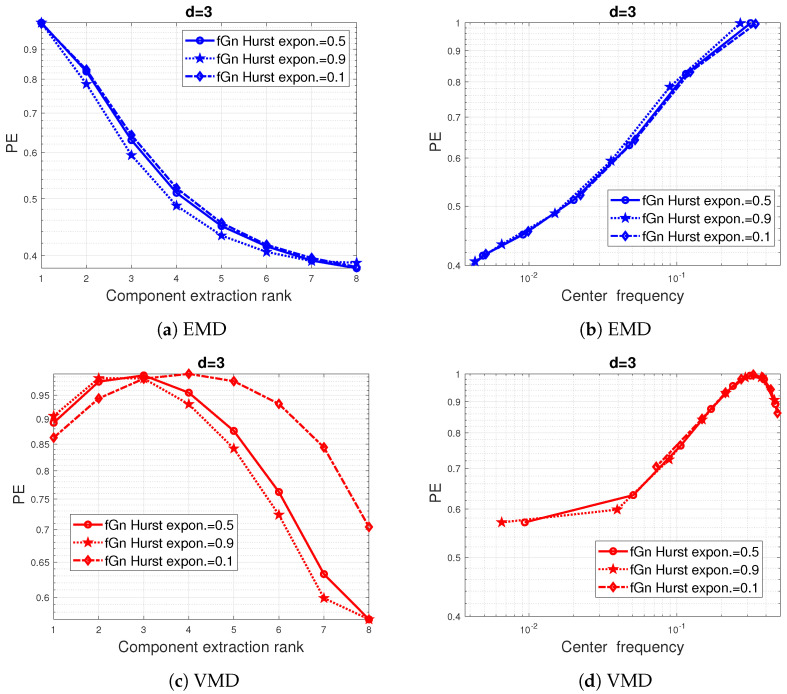
Impact of the data-driven decomposition on the PE values of white Gaussian noise (-o) compared to that of fractional Gaussian processes with Hurst exponent 0.1 (diamond) and 0.9 (star): (**a**,**b**) PE of a pattern dimension d=3 with EMD preprocessing. (**c**,**d**) same as (**a**,**b**) but with VMD preprocessing. A total of 100 times series were simulated for each Hurst exponent.

**Figure 6 entropy-24-01343-f006:**
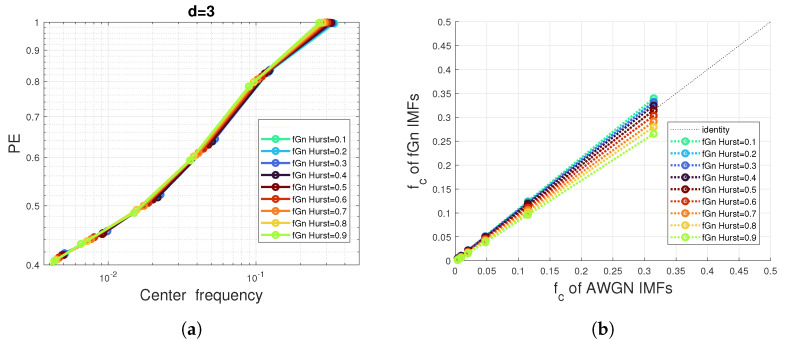
Impact of the EMD preprocessing on the PE values of white Gaussian noise (-o) versus fractional Gaussian processes: (**a**) PE of a pattern dimension d=3 and Hurst exponent varying from 0.1 to 0.9 in 0.1 steps. (**b**) Average of center frequencies of fGn IMFs versus average of center frequencies of AWGN IMFs. A total of 100 times series are simulated for each Hurst exponent.

**Figure 7 entropy-24-01343-f007:**
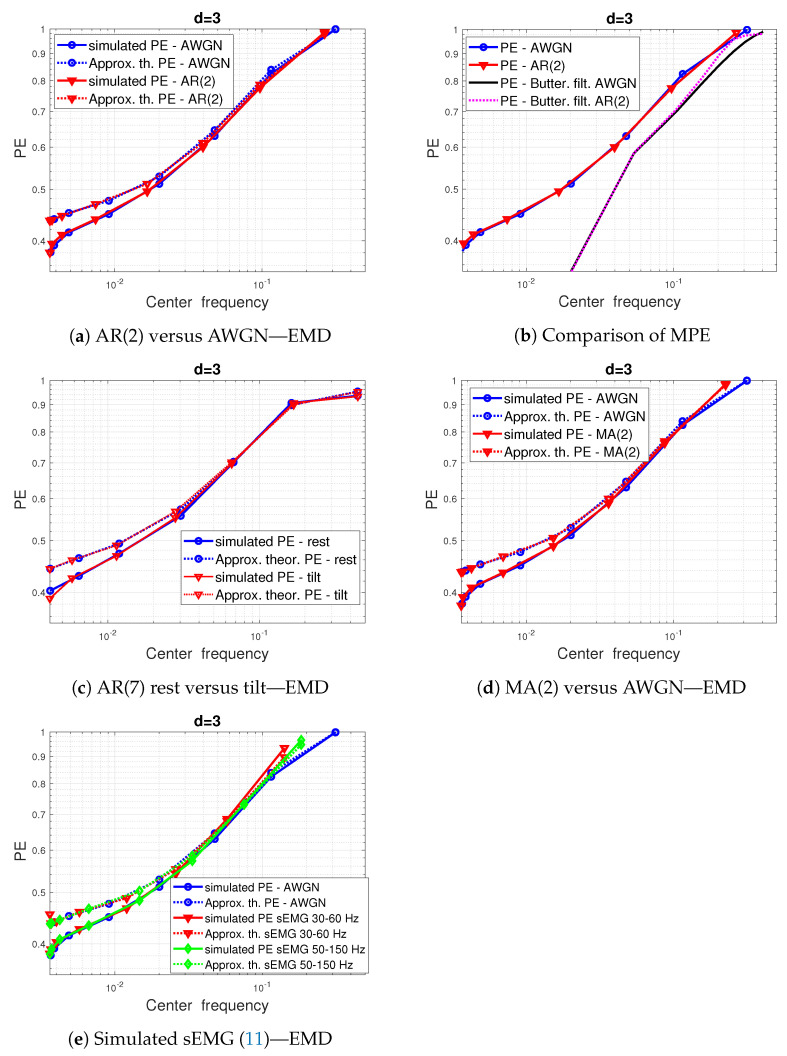
Impact of EMD decomposition on the PE values of signal models. (**a**) EMD-based MPE of AR(2) processes (-${}_{\Delta }$) versus white Gaussian noise(-o): experimental PE (solid line) versus theoretical PE approximation (dashed line). (**b**) EMD-based MPEs of both AR(2) model and AWGN superimposed to linear preprocessing-based MPE: 9th order Butterworth filter. (**c**) EMD-based MPE of AR(7) during rest and tilt positions: EMD-based MPE of AR(2) model (-${}_{\Delta }$) versus white Gaussian noise(-o): experimental PE (solid line) versus theoretical PE approximation (dashed line). (**d**) same as (**a**) but with MA(2) processes. (**e**) EMD-based MPE applied to two sEMG models (Equation 11).

**Figure 8 entropy-24-01343-f008:**
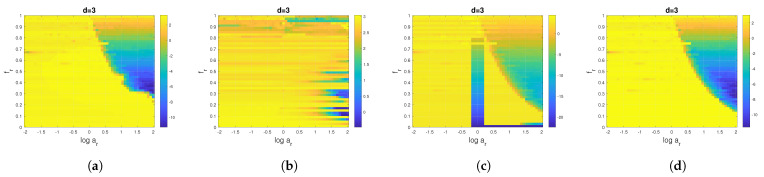
Impact of the data-driven decomposition on the PE values of a sum of two sinusoidal signals: the relative error between theoretical PE (Equation 17) in percent and computed PE of pattern dimension d=3 of the first component obtained after (**a**) EMD, (**b**) VMD, (**c**) EWT and (**d**) SSA decomposition.

**Figure 9 entropy-24-01343-f009:**
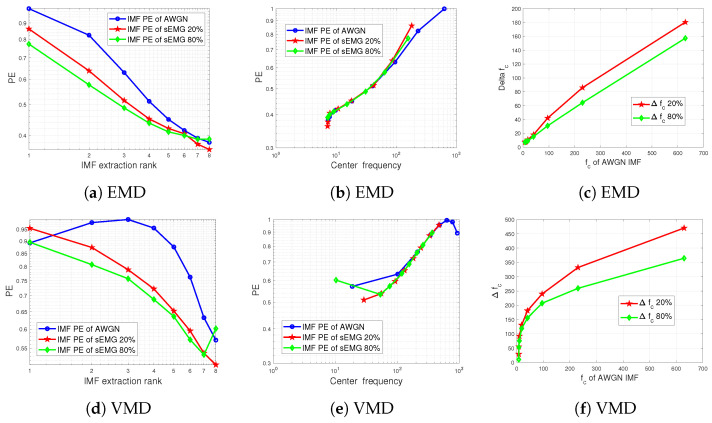
Impact of EMD and VMD decomposition on the PE values of real-life sEMG signals versus AWGN: (**a**,**c**) IMF PE of a pattern dimension d=3 as a function of IMF extraction rank. (**b**,**d**) IMF PE of a pattern dimension d=3 as a function of center frequencies of extracted IMFs. Center frequencies of extracted IMFs from sEMG signals versus those of AWGN.

## Data Availability

Not applicable.

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
