# Peer review of "On the Genuine Relevance of the Data-Driven Signal Decomposition-Based Multiscale Permutation Entropy"

_entropy, 2022, doi:10.3390/e24101343_

Round 1
Reviewer 1 Report
This study analyzed the non-linear scale PE of a signal decomposed by using EMD, VMD, EWT, and SSA. The results of the simulation and the approximate theory are also compared. Those results provided a correct interpretation when applying the PE to each component of the signal decomposed by EMD, VMD, EWT, and SSA. For the general readers, I would like to provide some questions and suggestions, as follows.- The EMD has some limitations which the authors described in the manuscript. However, which type of EMD authors applied in this study still remains unclear. As well, the mode mixing problem of the original EMD may lead to strange results of PE. The authors shall answer it completely.
- Why EWT and SSA were not applied after section 3.2?
- Most of the figures' results show that if the central frequencies are similar, the PE values will also be similar. Are there counter-examples?
- Since PE is a pattern-based entropy method, if possible, please provide the decomposition result (IMF1, IMF2, ..., IMFn) of EMD, VMD, EWT, and SSA. For the general readers, I expect the authors can compare the difference among the pattern of IMFs decomposed by different methods.
Author Response
Please find the answers to reviewer #1 in the attached file.

Reviewer 2 Report
The article is very well written and presents a novel approach to understand the impact of preprocessing steps on the interpretation of permutation entropy (PE) values computed by different types of signal decomposition methods.
The authors presented their approach for plotting PE values computed from different IMFs of time varying signals with respect to their normalized center frequency instead of plotting as a function of extraction rank of the components. This results in an outcome wherein all the data decomposition methods leads to almost similar variation of PE but with different bandwidths. The authors rightly point out that interpretation of signals for different application using PE values should take into account how different preprocessing steps influence the shift in bandwidth of IMFs.
The only thing that seems to be missing is using an array of real-life signals from different applications to further reinforce or buttress their argument that using mean frequency to plot PE values alter the interpretation of the results. The results presented in this paper from real life sEMG signals, and the ensuing discussion do not present a clear picture to the readers about the applicability or usefulness of the presented approach.
Author Response
Please find the answers to reviewer #2 in the attached file.
